# Differences in the Clinical Profile and Management of Atrial Fibrillation According to Gender. Results of the REgistro GallEgo Intercéntrico de Fibrilación Auricular (REGUEIFA) Trial

**DOI:** 10.3390/jcm10173846

**Published:** 2021-08-27

**Authors:** Olga Durán-Bobín, Juliana Elices-Teja, Laila González-Melchor, María Vázquez-Caamaño, Emiliano Fernández-Obanza, Eva González-Babarro, Pilar Cabanas-Grandío, Miriam Piñeiro-Portela, Oscar Prada-Delgado, Mario Gutiérrez-Feijoo, Evaristo Freire, Oscar Díaz-Castro, Javier Muñiz, Javier García-Seara, Carlos Gonzalez-Juanatey

**Affiliations:** 1Cardiology Department, Hospital Universitario Lucus Augusti e Instituto de Investigación Sanitaria de Santiago de Compostela (IDIS), 27002 Lugo, Spain; odbobin@gmail.com (O.D.-B.); julielices@gmail.com (J.E.-T.); 2Cardiology Department, Hospital Clínico de Santiago de Compostela, CIBERCV, 15706 A Coruña, Spain; dra_glezmelchor@hotmail.com (L.G.-M.); javiergarciaseara@yahoo.es (J.G.-S.); 3Cardiology Department, Hospital Povisa, 36211 Vigo, Spain; mcaamano@povisa.es; 4Cardiology Department, Hospital Arquitecto Marcide, 15405 Ferrol, Spain; windscheid@hotmail.com; 5Cardiology Department, Hospital Montecelo, 36162 Pontevedra, Spain; egbabarro@hotmail.com; 6Cardiology Department, Hospital Álvaro Cunqueiro e Instituto de Investigación Sanitaria Galicia Sur (IISGS), 36312 Vigo, Spain; pilicgrandio@yahoo.es (P.C.-G.); oscar.diaz.castro@sergas.es (O.D.-C.); 7Cardiology Department, Complexo Hospitalario Universitario de A Coruña, Instituto de Investigación Biomédica de A Coruña (INIBIC), 15006 A Coruña, Spain; miriam.pineiro.portela@sergas.es (M.P.-P.); oscar.prada.delgado@sergas.es (O.P.-D.); 8Cardiology Department, Hospital de Ourense, 32616 Ourense, Spain; mgutfei@gmail.com (M.G.-F.); efreirec@telefonica.net (E.F.); 9Grupo de Investigación Cardiovascular, Departamento de Ciencias de las Salud e Instituto de Investigación Biomédica de A Coruña (INIBIC), Universidade da Coruña, CIBERCV, 15006 A Coruña, Spain; jamu@udc.es

**Keywords:** atrial fibrillation, gender, anticoagulation, rhythm control

## Abstract

To analyze the clinical profile and therapeutic strategy in atrial fibrillation (AF) according to gender in a contemporaneous patient cohort a prospective, multicenter observational study was performed on consecutive patients diagnosed with AF and assessed by cardiology units in the region of Galicia (Spain). A total of 1007 patients were included, of which 32.3% were women. The mean age of the women was significantly greater than that of the men (71.6 versus 65.7 years; *p* < 0.001), with a higher prevalence of hypertension (HTN) and valve disease. Women more often reported symptoms related to arrhythmia (28.2% in EHRA class I versus 36.4% in men), with a poorer level of symptoms (EHRA classes IIb and III). Thromboembolic risk was significantly higher among women (CHA_2_DS_2_-VASc 3 ± 1.3 versus 2 ± 1.5), in the same way as bleeding risk (HAS-BLED 0.83 ± 0.78 versus 0.64 ± 0.78) (*p* < 0.001), and women more often received anticoagulation therapy (94.1% versus 87.6%; *p* = 0.001). Rhythm control strategies proved significantly less frequent in women (55.8% versus 66.6%; *p* = 0.001), with a lesser electrical cardioversion (ECV) rate (18.4% versus 27.3%; *p* = 0.002). Perceived health status was poorer in women. Women were older and presented greater comorbidity than men, with a greater thromboembolic and bleeding risk. Likewise, rhythm control strategies were less frequent than in men, despite the fact that women had poorer perceived quality of life and were more symptomatic.

## 1. Introduction

Atrial fibrillation (AF) is the most common sustained heart arrhythmia in the general population [1,2], with a current prevalence of 2–4% [1,3,4,5] that is expected to more than double in the coming years due to the prolongation of life expectancy and the development of new detection methods [6].

There is evidence of gender-related differences in relation to AF, though women are under-represented in the existing studies [2]. In this respect, it has been reported that the age-adjusted incidence, prevalence and risk of suffering AF during life are lower in women than in men [2]. Furthermore, women are older at the time of detection of the arrhythmia, and the risk factor and comorbidity profiles differ with respect to men, in the same way as the symptoms burden [2]. Women present a higher CHA_2_DS_2_-VASc score, and stroke among females tends to be more severe and disabling [7]. However, there are data suggesting that women present poorer control of anticoagulation with warfarin than men, with a greater residual risk despite correct control with vitamin K antagonists (VKAs] [7]. The efficacy of direct-acting oral anticoagulants (DOACs) does not appear to vary according to gender, though here again women are under-represented in the pivotal studies [7].

The clinical management strategy also differs between men and women. The available data indicate more frequent prescription of rate control strategies in women [2]. On the other hand, the use of antiarrhythmic drugs (AADs) in women is associated to a greater incidence of prolonged QT-interval arrhythmic events with class IA and III drugs, and of sinus node dysfunction requiring pacemaker implantation [7].

The REgistro GallEgo Intercéntrico de Fibrilación Auricular (REGUEIFA) trial is a prospective, multicenter observational registry of consecutive patients diagnosed with AF [8]. The present study was carried out to analyze and compare the clinical characteristics of patients with AF and the therapeutic strategies used according to gender in the cohort of the REGUEIFA trial.

## 2. Material and Methods

The study design and methods have been previously described [8]. A total of 1007 patients were recruited between 2 January 2018 and 27 February 2020, belonging to the cohort of the REGUEIFA trial [8].

### 2.1. Inclusion Criteria

 − Men and women aged ≥ 18 years. − An electrocardiographic or external or implantable Holter monitoring diagnosis of AF with a duration of over 30 s. − A registered AF episode in the last year. − Possible sinus rhythm at the time of inclusion.

### 2.2. Exclusion Criteria

 − Patients in which long-term follow-up is not contemplated or not possible. − Patients with secondary transient AF due to reversible causes. − Patients enrolled in interventional studies conditioning treatments, frequency of consultation or diagnostic procedures.

### 2.3. Ethical Statement

The study protocol was performed according to the principles of the Helsinki Declaration and the Clinical Research Ethics Committee (CREC) of Galicia (Spain) approved the study, with reference code 2016/376.

### 2.4. Participating Hospitals

A total of eight hospitals of the Galician Health Department (Servicio Gallego de Salud (SERGAS)) participated in the study, covering 100% of the population in the region of Galicia (Spain).

#### 2.4.1. Conduct of the Study

All the patients meeting the inclusion criteria and none of the exclusion criteria were considered for enrollment in the study, and the corresponding data were entered in the electronic case report form (eCRF). The collected information was sent to the registry coordinating center through secure websites, with due observation of data confidentiality. All patients received the patient information sheet, and written informed consent was obtained in accordance with the local requirements before any of the study-related procedures were carried out (i.e., data transfer from the case histories to the eCRF). All patient-related information complied with Act 3/2018, of 5 December, referred to personal data protection, and with Act 41/2002, regulating patient autonomy and rights and obligations in relation to clinical documentation and information. All consenting patients were assigned a specific number code serving as personal identifier.

#### 2.4.2. Data Registry

The study data were collected on occasion of the baseline visit. The data referred to the international normalized ratio (INR) of those patients receiving treatment with VKAs were also recorded. The information was entered in the eCRF designed by the principal investigators and produced by the company Odds S.L. (Odds S.L., A Coruña, Spain)

#### 2.4.3. Quality Assurance and Control

The study supervisor checked 10% of all the eCRF against the source documentation in the study centers, in accordance with the supervision protocol.

### 2.5. Statistical Analysis

Quantitative and qualitative variables were reported as the mean ± standard deviation (SD) and percentages, respectively. The differences between the two groups (men versus women) were analyzed using the chi-square test or Fisher exact test in the case of qualitative variables, and the Mann-Whitney U-test (two-sample Wilcoxon rank-sum test) in the case of quantitative variables. The Kruskal-Wallis test was used to compare the INR levels in the last 6 months and the time in therapeutic range (TTR) between men and women.

Statistical significance was considered for *p* = 0.05.

## 3. Results

### 3.1. Basal Characteristics

#### Cardiovascular Risk Factors (CVRFs)

Thirty-two percent of the patients included in the study were women. The mean age of the global sample was 67.6 years, and HTN was the most prevalent CVRF (62%). Compared with the men, the females in our study were significantly older (71.6 versus 65.7 years on average; *p* < 0.001) and were more often hypertensive (*p* = 0.05). No gender differences were observed in relation to other risk factors such as obesity, diabetes mellitus or dyslipidemia. Toxic habits (smoking and alcohol) were more common in men (*p* < 0.001) (Table 1).

### 3.2. Complementary Tests

Creatinine concentration was significantly lower in women (*p* < 0.01). In turn, the left ventricular ejection fraction (LVEF) was average 4% higher than in men (*p* < 0.01). The diameter of the left atrium was significantly smaller in women, though not so the atrial volume.

Nine percent of the patients presented bundle block, with a significant two-fold higher prevalence in men versus women (10% versus 5%; *p* < 0.01). Right bundle block was the most frequent presentation. Both left and right bundle block were more prevalent in men, though statistical significance was not reached (*p* = 0.19 and *p* = 0.06 for left and right bundle block, respectively) (Table 2).

### 3.3. Cardiovascular History

Approximately 15% of the patients had a history of heart failure (HF). There were no gender differences in the prevalence of HF or in NYHA functional class. Coronary disease was almost twice as frequent in men versus women—the difference being statistically significant (13.3% versus 7.6%; *p* = 0.008). A history of myocardiopathy was likewise approximately two times more common in men (9.4% versus 4.2%; *p* = 0.005). In contrast, valve disease was more frequent in women (14.7% versus 10.1%; *p* = 0.03). In turn, almost 10% of the patients carried a pacemaker or implantable cardioverter defibrillator (ICD), with no differences between men and women. Pacemakers were the most frequent devices in both genders. On the other hand, women reprsented only three of the total of 24 ICDs in the global study sample (Table 3).

### 3.4. Personal History, Comorbidities and Previous Thromboembolic Events and Bleeding Episodes

The prevalence of chronic obstructive pulmonary disease (COPD) was 50% lower in women than in men (6.4% versus 12.7%; *p* = 0.002), while hypothyroidism was almost four times more frequent in women (11.9% versus 3.9%; *p* < 0.001). There were no gender differences in terms of the presence of hyperthyroidism or neoplastic disease. Six percent of the patients had suffered thromboembolic events, with stroke representing almost half of the total (2.8%). In turn, 3.5% of the study population had a history of bleeding, with no differences between men and women (Table 4).

### 3.5. Risk Scales

The mean CHA_2_DS_2_-VASc risk score was 3 in women and 2 in men (*p* < 0.001). The mean HAS-BLED bleeding risk score was also slightly higher in women (0.8 versus 0.6; *p* < 0.001) (Table 5).

### 3.6. Characteristics of Atrial Fibrillation

Most patients (38.9%) presented level IIa of the EHRA symptoms classification. The percentage of asymptomatic individuals (level I of the EHRA classification) was higher in the male group (36.4% versus 28.2%), while women more often presented class II and class III (19.6% versus 16% and 11.3% versus 8.5%, respectively) (Table 6).

Statistically significant differences were also observed in terms of the clinical type of AF (*p* = 0.009). In this regard, a distinction was made between first diagnosis AF and paroxysmal, persistent, long duration persistent and permanent AF. Persistent AF was the most frequent presentation both globally (26.3%) and in men (29.3%). In contrast, permanent AF was the most common presentation in women (29.7%). There were no significant gender differences in relation to paroxysmal AF (*p* = 0.1) (Table 6).

### 3.7. Previous Treatment Strategy (in Non-First Diagnosis of AF)

Previous rhythm control had been decided in 58.6% of the women and in 67.3% of the men (*p* = 0.03). Electrical cardioversion had been performed in 59% of the men and in 50% of the women (*p* = 0.09). Twenty-one percent of the men had undergone pulmonary vein ablation versus 13.9% of the women (*p* = 0.08). In turn, atrioventricular node ablation had been performed in three patients, of which two were women (Table 6).

### 3.8. Current Treatment Strategy

Rhythm control strategies were being applied in 66.6% of the men and in 55.8% of the women—the difference being statistically significant (*p* = 0.001). Electrical cardioversion was more often performed in men (27.3% versus 18.4%; *p* = 0.002). There were no gender differences in AF catheter ablation (16.8% versus 16.2%; *p* = 0.8). On the other hand, atrioventricular node ablation remained more frequent in women (2% versus 0.2%; *p* = 0.007) (Table 7).

Antiarrhythmic drugs were prescribed in 42.2% of the men and in 31.2% of the women (*p* = 0.001). Amiodarone, followed by flecainide, were the most frequently used AADs. The distribution of the use of the five drugs contemplated in the study was similar in both groups.

No gender differences were observed in terms of the use of β-blockers, calcium antagonists or dihydropyridinic agents, digoxin, angiotensin II converting enzyme inhibitors (ACEIs) or angiotensin II receptor antagonists (ARAs), statins or antiplatelet drugs, or as regards the administration of dual therapy (antiplatelet medication and anticoagulation). In contrast, diuretics were more often used by women (40.4% versus 31.4%; *p* = 0.005) (Table 8).

On comparing the basal characteristics of the women and men offered rhythm control, the former was seen to be older (67.9 versus 60.9 years; *p* < 0.001), with a smaller left atrium (diameter 42 mm versus 44.8 mm, *p* = 0.002; volume 42.5 mL/m^2^ versus 55.3 mL/m^2^, *p* = 0.054), and a significantly greater LVEF (59.3% versus 55.2%; *p* = 0.002). Women also presented paroxysmal AF more often (39% versus 28.4%; *p* = 0.009), with more severe symptoms according to the EHRA classification (*p* = 0.027). Likewise, women presented a higher CHA_2_DS_2_-VASc score (2.3 versus 1.5; *p* < 0.001), were more often hypertensive (*p* = 0.1), and had a lesser incidence of COPD (3.8% versus 8.1%; *p* = 0.058) (Table 9).

### 3.9. Anticoagulation

Almost 90% of the global patients received anticoagulation–the proportion being higher in women (94.1% versus 87.6%; *p* = 0.001). Vitamin K antagonists were the most widely prescribed drugs (58.8%), followed by dabigatran (34.4%), rivaroxaban (28.2%), apixaban (23.9%) and edoxaban (14.4%). The mean time in therapeutic range was 54.8%, with no statistically significant differences between men and women. The use of direct-acting oral anticoagulants (DOACs) was more common in men (43.8% versus 35.8%, *p* = 0.2). Dabigatran was the most widely used drug, while the second most frequently prescribed drug was rivaroxaban in men and apixaban in women (Table 8).

### 3.10. EQ-5D and ACTS Questionnaires

The EuroQol-5D (EQ-5D) questionnaire was used to assess health-related quality of life. This tool comprises five health dimensions (mobility, personal care, daily activities, pain/discomfort and anxiety/depression), each of which has three levels of severity (no problems, some problems or moderate problems, and serious problems). Statistically significant gender differences were recorded (*p* < 0.001) terms of the EQ-5D (summarizing score), with higher scores in men (0.85 versus 0.74) (Table 10).

The index ranges from 1 to 0. On the other hand, the mean EQ-5D score was seen to be higher among the patients subjected to rhythm control than in those in which rate control was decided, regardless of gender–though both management strategies were more commonly used in men (Figure 1).

The anti-clot treatment scale (ACTS) was used as a specific satisfaction score to assess burden (higher scores reflecting lesser burden) and benefit (higher scores reflecting greater benefit) referred to anticoagulation therapy. Men reported greater satisfaction with the treatment (burden score 53.9 versus 52.3; *p* < 0.002). However, no gender differences were observed in terms of the benefit score, negative impact (general burden score), or positive impact (general benefit score) referred to anticoagulant drug use (Table 11).

## 4. Discussion

Our results suggest that once the different characteristics of men and women are considered, there are no differences between the sexes in the level of anticoagulation or goals achieved, while differences are observed in the therapeutic strategy of AF that may have implications for the quality of life of these patients.

In our study, proximately one-third of the patients in our study (32.3%) were women. This smaller proportion of females is consistent with the data found in the literature, where lower incidences and prevalences are reported, together with a lesser risk of developing AF during life in comparison with males [2]. In North America and Europe, the age-adjusted incidence of AF is 1.5–2 times higher in men than in women, as evidenced by the Framingham trial [3,4,5] and the Olmstead County trial in Minnesota [9]. Furthermore, a disproportionate increase in the incidence of AF has been observed with aging in both genders [10]. The age-adjusted prevalence was also found to be lower in women (7.4% versus 10.3% in men) in a retrospective study of Medicare patients in the United States aged 65 years and older. These figures in turn are similar to those reported by a randomized trial in Sweden in 75-year-old subjects (9.2% in women versus 15% in men) [11].

In the last 50 years we have evidenced a rise in both the incidence and prevalence of AF in the population [1,2], with no significant differences in the growth rates between males and females. Nevertheless, the absolute number of women with AF is greater than the number of men, due to the longer life expectancy of the former [12]. In our study, the women were 6 years older than the men on average (71.6 versus 65.7 years; *p* < 0.001). This observation is consistent with the data found in the literature, where in general women with AF are older than the men [3]. Age appears to be the most important risk factor for AF, with an up to two-fold increase in incidence for every additional 10 years of age [13]. The results of the BiomarCaRE consortium [12] indicate that on average, women develop AF a decade later than men. A probable hormone protective effect has been postulated in this regard, making development of the arrhythmia unlikely before menopause [14].

In concordance with the observations of other studies our women presented a higher prevalence of HTN and valve disease than the men, and a lower prevalence of coronary disease [15]. On the other hand, males had a significantly higher alcohol intake than women, and tended to present a greater association between alcohol consumption and the risk of developing AF [16].

COPD, strongly associated to smoking, was significantly more frequent in men (46% of whom were smokers, versus 11% of the women), and was associated to the development of AF [17].

From the structural perspective, we know that dilatation of the left atrium is associated to an increased risk of AF [18]. In this regard, the left atrium was generally smaller in our women (the mean diameter being 42.7 mm versus 44.5 mm in males), which could contribute to explain the lesser prevalence of AF among females.

In addition to being associated to an increase in cardiovascular events and mortality [19], we know that AF has a negative impact upon patient quality of life [20]. In our study, women reported arrhythmia-related symptoms more often (28.2% of women in class I versus 36.4% of men), and moreover presented a poorer level of symptoms (EHRA classes IIb and III). The analysis of the database of the Euro Observational Research Program on Atrial Fibrillation (EORP-AF) Pilot survey [17], which examined the gender-related differences among patients with AF in Europe, showed women to be more often symptomatic than men, with a greater proportion of women in EHRA classes III and IV (*p* = 0.0012).

Permanent AF was the most frequent presentation of arrhythmia in women, with no significant differences in the prevalence of paroxysmal AF between the two genders. This is in contrast to the findings of the EORP-AF trial [17], where the prevalence of paroxysmal AF was seen to be higher in females (28.5% versus 25.1%), with very similar permanent AF rates in both groups (17.5% and 17.1%). Likewise, the BEAT-AF [21], a prospective, multicenter observational study of 1553 patients with AF (mean age 70 ± 11 in women and 67 ± 12 years in men), reported a higher frequency of paroxysmal AF in women (60.6% versus 53.7%; *p* = 0.04). In further considering gender-related differences, a multivariate analysis found the female gender to remain as a potent predictor of symptoms (odds ratio [OR]: 2.6; 95% confidence interval [95%CI], 2.1–3.4; *p* < 0.0001), adjusted for variables such as age or the presence of comorbidities, within a sub-study of the ORBIT AF registry [22].

In our study, women had poorer perceived health as determined by the EQ-5D. This is consistent with the observations of the BEAT-AF study [21] and EORP-AF trial [17]. In both genders, perceived health status was better in those patients subjected to a rhythm control management strategy versus rate control, though the scores were comparatively higher in men. Other studies have evidenced that if sinus rhythm is achieved and maintained, similar improvement of quality of life is observed in both genders [23]. In the BATE-AF trial, the symptoms burden decreased 56% among those patients subjected to rhythm control, but only 28.4% in the absence of such treatment [21].

Women present symptoms more often than men, though a comparatively lesser proportion of them are subjected to rhythm control measures compared with men. rate control is more common, despite the fact that it seems clear that rhythm control strategies improve quality of life and perceived health status, independently of patient gender [1,23].

A total of 55.8% of the women in our study were subjected to rhythm control versus 66.6% of the men (*p* = 0.001), with statistically significant differences in the electrical cardioversion (ECV) rate, which proved lower in women (27.3% versus 18.4%; *p* = 0.002). No gender differences were observed in relation to catheter ablation (16.8% versus 16.2%; *p* = 0.8). These observations are similar to those of the EORP-AF trial [17], with ECV rates in women and men of 18.9% versus 25.5%, respectively (*p* = 0.0001). On stratifying according to symptoms, in the presence of typical symptoms, women were less likely to be subjected to rhythm control strategies (*p* = 0.002). On the other hand, the women in the BEAT-AF study [21] had a significantly lower prevalence of interventions (understood as ECV or ablation) than the men (30.7% versus 38.9%; *p* = 0.002).

It could be speculated whether this lower use of rhythm control strategy in the women of our study was due to the clinical profile of the women who developed AF (age, comorbidities, etc.), the perception of a greater frequency of complications with the invasive strategy (with the reporting of a risk of complications 1.28 to 2.3 times higher than in men) [24], lesser efficacy (women have greater atrial fibrosis and a greater presence of triggers outside the pulmonary veins) [25], or the increased risk of malignant ventricular arrhythmias associated to the use of class III AADs [26,27]. In our study, men received AADs more often than women, in clear correlation to the greater use of rhythm control strategies among the former. In the patients of the EORP-AF study [17], the differences were less manifest (39.7% versus 36.6%) and failed to reach statistical significance.

Although we recorded no statistically significant differences in the prevalence of thromboembolic events between men and women, the latter presented greater risk, since the CHA_2_DS_2_-VASc risk score, with a mean value of 3, was one point higher than in men (*p* < 0.001). The HAS-BLED bleeding risk score was also higher in women (0.8 versus 0.6; *p* < 0.001). In this regard, women more often received anticoagulation therapy (94.1% versus 87.6%; *p* = 0.001), in abidance with the clinical practice guides, which describe the female gender as an independent predictor of stroke risk in the presence of one or more risk factors for thromboembolism [7]. The results of the EORP-AF study showed oral anticoagulation to be used in a similar manner in both genders (81.5% versus 79.8%; *p* = 0.36), with VKAs being the most commonly prescribed drugs (72.7%). On considering a CHA_2_DS_2_-VASc risk score of ≥2, the differences were found to be significant in favor of women (94.7% versus 74.6%; *p* < 0.0001). Vitamin K antagonists were also the most frequently used drugs in the patients of our study (58.8%). The mean time in therapeutic range was 54.8%, with no statistically significant differences between men and women. The use of DOACs was more frequent in men (43.8% versus 35.8%), though statistical significance was not reached (*p* = 0.2).

## 5. Conclusions

The analysis of the patients in the REGUEIFA study revealed differences between men and women diagnosed with AF in Galicia. Women were older and presented greater comorbidity compared with men. Due to the comparatively greater thromboembolic risk in women, the latter received anticoagulation treatment more often than men, in concordance with the clinical practice guides, with no differences regarding the time in therapeutic range between the two genders. In general, rhythm control measures were less frequently prescribed in women than in men, despite the fact that the former had poorer perceived quality of life and were more often symptomatic. Knowing that rhythm control improves the quality of life of patients of either gender, we should seek to improve the treatment and clinical management of women with AF.

## Figures and Tables

**Figure 1 jcm-10-03846-f001:**
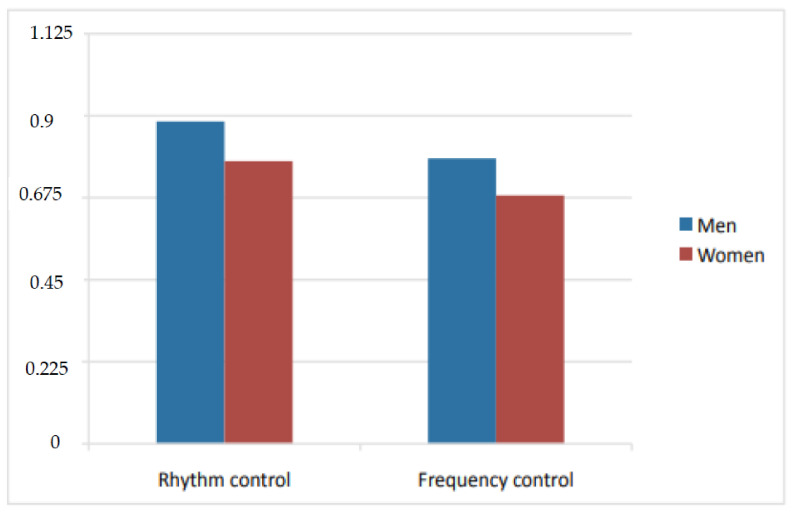
Mean EQ-5D score in men and women according to the treatment strategy used (rhythm control versus frequency control).

**Table 1 jcm-10-03846-t001:** Basal characteristics.

**Variables**		**Women RhC**	**Men RhC**	**Women RC**	**Men RC**	***p*** **-Value**
Age	*n*	182	454	144	227	<0.001
	mean ± SD	67.94 ± 8.96	60.98 ± 10.91	76.44 ± 8.57	75.22 ± 9.68	
Body mass index	*n*	181	452	144	227	0.955
	mean ± SD	29.92 ± 5.27	29.73 ± 4.74	29.18 ± 5.83	29.44 ± 4.79	
Diameter of LA (mm)	*n*	87	233	80	124	0.002
	mean ± SD	42.07 ± 8.92	44.86 ± 10.8	43.44 ± 12.77	43.93 ± 14.07	
Volume of LA (ml/m2)	*n*	33	87	30	55	0.054
	mean ± SD	42.51 ± 19.96	55.3 ± 34.86	65.98 ± 38.42	62.47 ± 30.44	
LVEF (%)	*n*	117	281	102	167	0.002
	mean ± SD	59.32 ± 10.71	55.26 ± 12.4	58.75 ± 11.19	53.5 ± 14.64	
HF based on NYHA functional class (*n*)		17	64	29	40	0.009
I		0.00%	28.13%	17.24%	10.00%	
II		58.82%	54.69%	58.62%	50.00%	
III		41.18%	17.19%	20.69%	40.00%	
IV		0.00%	0.00%	3.45%	0.00%	
CHA_2_DS_2_-VASc scale	*n*	182	454	144	227	<0.001
	mean ± SD	2.62 ± 1.32	1.52 ± 1.32	3.64 ± 1.22	3 ± 1.35	
HTN	*n*	182	454	144	227	0.101
	%	61.54	54.41	72.92	71.81	
COPD	*n*	182	454	144	227	0.058
	%	3.85	8.15	9.72	22.03	
Paroxysmal AF	*n*	182	454	144	227	0.009
	%	39.01	28.41	7.64	5.29	
EHRA classification (*n*)		182	454	144	227	0.027
I		22.53%	33.70%	35.42%	41.85%	
II a		46.70%	42.07%	32.64%	30.40%	
II b		20.33%	16.52%	18.75%	14.98%	
III		10.44%	6.83%	12.50%	11.89%	
IV		0.00%	0.88%	0.69%	0.88%	

AF: atrial fibrillation; COPD: chronic obstructive pulmonary disease; EHRA: European heart rhythm association; HTN: hypertension; LA: left atrium; NYHA: New York Heart Association; RhC: rhythm control; RC: rate control; HF: heart failure; LVEF: left ventricular ejection fraction.

**Table 2 jcm-10-03846-t002:** Complementary tests.

Variables	All		Men		Women		*p*-Value
Complementary tests							
Creatinine (mg/dL)	956	0.99 ± 0.36	646	1.04 ± 0.35	310	0.88 ± 0.34	<0.001
LVEF (%)	667	56.06 ± 12.73	448	54.6 ± 13.29	219	59.05 ± 10.91	<0.001
Diameter of LA (mm)	524	43.96 ± 11.7	357	44.54 ± 12.02	167	42.73 ± 10.92	0.011
Volume of LA (mL/m^2^)	205	56.73 ± 32.94	142	58.07 ± 33.29	63	53.68 ± 32.18	0.212
Bundle block	90	8.94%	71	10.43%	19	5.83%	0.017
LBB	38/90	42.22%	27/71	38.03%	11/19	57.89%	0.190
RBB	51/90	56.67%	44/71	61.97%	7/19	36.84%	0.068

LVEF: left ventricular ejection fraction; LA: left atrium; LBB: left bundle block; RBB: right bundle block.

**Table 3 jcm-10-03846-t003:** Cardiovascular history.

Variables	All		Men		Women		*p*-Value
Cardiovascular history							
Type of device	100		72		28		0.068
Pacemaker carrier	76	76.00%	51	70.83%	25	89.29	
ICD carrier	24	24.00%	21	29.17%	3	10.71	
Heart failure	150	14.90%	104	15.27%	46	14.11	0.628
HF based on NYHA functional class	150		104		46		0.213
I	27	18.00%	22	21.15%	5	10.87%	
II	82	54.67%	55	52.88%	27	58.70%	
III	40	26.67%	27	25.96%	13	28.26%	
IV	1	0.67%	0	0.00%	1	2.17%	
NYHA functional class ≥ II	123/150	82.00%	82/104	78.85%	41/46	89.13%	0.168
Coronary disease (ischemic heart disease)	116	11.52%	91	13.36%	25	7.67%	0.008
Valve disease	117	11.62%	69	10.13%	48	14.72%	0.033
Cardiomyopathy	78	7.75%	64	9.40%	14	4.29%	0.005

ICD: implantable cardioverter defibrillator; HF: heart failure. NYHA: New York Heart Association.

**Table 4 jcm-10-03846-t004:** Concomitant diseases.

Variables	All		Men		Women		*p*-Value
Personal history-concomitant diseases							
COPD	108	10.72%	87	12.78%	21	6.44%	0.002
Neoplasms	83	8.24%	56	8.22%	27	8.28%	0.975
Hyperthyroidism	19	1.89%	15	2.20%	4	1.23%	0.334
Hypothyroidism	66	6.55%	27	3.96%	39	11.96%	<0.001
Previous thromboembolic and bleeding events							
Thromboembolic events	61	6.06%	42	6.17%	19	5.83%	0.833
Ischemic stroke	29	2.88%	19	2.79%	10	3.07%	0.805
Bleeding events	36	3.57%	23	3.38%	13	3.99%	0.625

COPD: chronic obstructive pulmonary disease.

**Table 5 jcm-10-03846-t005:** Thromboembolic and bleeding risk scores.

Variables	All		Men		Women		*p*-Value
Thromboembolic/bleeding risk							
CHA_2_DS_2_-VASc scale	1007	2.35 ± 1.54	681	2.01 ± 1.5	326	3.07 ± 1.37	<0.001
HAS-BLED scale	1007	0.7 ± 0.78	681	0.64 ± 0.78	326	0.83 ± 0.78	<0.001

**Table 6 jcm-10-03846-t006:** Characteristics of previous atrial fibrillation management strategies.

Variables	All		Men		Women		*p*-Value
Characteristics of atrial fibrillation							
EHRA classification							0.052
I	340	33.76%	248	36.42%	92	28.22%	
IIa	392	38.93%	260	38.18%	132	40.49%	
IIb	173	17.18%	109	16.01%	64	19.63%	
III	95	9.43%	58	8.52%	37	11.35%	
IV	7	0.70%	6	0.88%	1	0.31%	
Clinical type of AF							0.009
First diagnosis	231	22.94%	160	23.49	71	21.78%	
Paroxysmal	223	22.14%	141	20.70	82	25.15%	
Persistent	265	26.32%	200	29.37%	65	19.94%	
Long duration persistent	26	2.58%	15	2.20%	11	3.37%	
Permanent	262	26.02%	165	24.23%	97	29.75%	
Previous strategy (if not first diagnosis of AF)							
Rhythm control	403/627	64.27%	274/407	67.32%	129/220	58.64%	0.030
Electrical cardioversion	403		274		129		0.171
No	176	43.67%	112	40.88%	64	49.61%	
1	143	35.48%	99	36.13%	44	34.11%	
>1	84	20.84%	63	22.99%	21	16.28%	
Electrical cardioversion (yes/no)	227/403	56.33%	162/274	59.12%	65/129	50.39%	0.099
Ablation of AF	403		274		129		0.089
No	324	80.40%	215	78.47%	109	84.5%	
Isolation of pulmonary veins	76	18.86%	58	21.17%	18	13.95%	
AV node ablation	3	0.74%	1	0.36%	2	1.55%	

AF: atril fibrillation; AV: atrioventricular; EHRA: European heart rhythm association.

**Table 7 jcm-10-03846-t007:** Current clinical management strategies.

Variables	All		Men		Women		*p*-Value
Current treatment of atrial fibrillation							
Rhythm control strategy	636	63.16%	454	66.67%	182	55.83%	0.001
Catheter ablation	168	16.68%	115	16.89%	53	16.26%	0.802
Number of interventions	168	1.14 ± 0.38	115	1.15 ± 0.38	53	1.11 ± 0.38	0.435
Electrical cardioversion	246	24.43%	186	27.31%	60	18.4%	0.002
Number of interventions	246	1.32 ± 0.76	186	1.36 ± 0.83	60	1.18 ± 0.47	0.227
Implantation of devices	38	3.77%	22	3.23%	16	4.91%	0.191
Rate control interventions AV node ablation	9	0.89%	2	0.29%	7	2.15%	0.007

AV: atrioventricular.

**Table 8 jcm-10-03846-t008:** Pharmacological treatment.

Variables	All		Men		Women		*p*-Value
Medication							
Antiarrhythmic treatment	**390**	38.73%	288	42.29%	102	31.29%	0.001
Group I and/or III antiarrhythmic drugs	389/390	99.74%	287/288	99.65%	102/102	100.00%	1.000
Type of group I and/or III antiarrhythmic drugs	389		287		102		0.321
Amiodarone	182	46.79%	129	44.95%	53	51.96%	
Dronedarone	6	1.54%	3	1.05%	3	2.94%	
Sotalol	13	3.34%	9	3.14%	4	3.92%	
Flecainide	183	47.04%	142	49.48%	41	40.20%	
Propafenone	5	1.29%	4	1.39%	1	0.98%	
Beta-blockers	689	68.42%	464	68.14%	225	69.02%	0.778
Non-dihydropyridinic calcium antagonists	54	5.36%	37	5.43%	17	5.21%	0.886
Digoxin	62	6.16%	37	5.43%	25	7.67%	0.167
ACEIs/ARAs	517	51.34%	350	51.40%	167	51.23%	0.960
Diuretics	346	34.36%	214	31.42%	132	40.49%	0.005
Statins	466	46.28%	317	46.55%	149	45.71%	0.802
Antiplatelet medication	43	4.27%	33	4.85%	10	3.07%	0.192
Anticoagulation + antiplatelet drugs	27	2.68%	21	3.08%	6	1.84%	0.302
Anticoagulation	904	89.77%	597	87.67%	307	94.17%	0.001
Type of anticoagulant	904		597		307		0.020
Vitamin K antagonist	532	58.85%	335	56.11%	197	64.17%	
DOACs	372	41.15%	262	43.89%	110	35.83%	
Type of DOAC	372		262		110		0.249
Dabigatran	128	34.41%	91	34.73%	37	33.64%	
Edoxaban	50	13.44%	35	13.36%	15	13.64%	
Rivaroxaban	105	28.23%	80	30.53%	25	22.73%	
Apixaban	89	23.92%	56	21.37%	33	30.00%	

ACEIs: angiotensin II converting enzyme inhibitors; ARAs: angiotensin II receptor antagonists; DOACS: direct-acting oral anticoagulants. 390 patients under Antiarrhythmic treatment (in bold).

**Table 9 jcm-10-03846-t009:** Basal characteristics of women and men subjected to rhythm control (RhC) versus rate control (RC).

Variables	All		Men		Women		*p*-Value
	*n*		*n*		*n*		
Patients	1007		681	67.63	326	32.37	
Age	1007	67.66 ± 11.98	681	65.73 ± 12.47	326	71.69 ± 9.74	<0.001
Body Mass Index (BMI)	1004	29.62 ± 5.02	679	29.63 ± 4.75	325	29.59 ± 5.53	0.405
Cardiovascular Risk Factors							
Tobacco							<0.001
Newer	658	65.34%	369	54.19%	289	88.65	
Current smoker	86	8.54%	75	11.01%	11	3.37%	
Recent ex-smoker (< 6 months)	19	1.89%	19	2.79%	0	0.00%	
Former smoker (≥ 6 months)	244	24.23%	218	32.01%	26	7.98%	
Smokes (yes)	86	8.54%	75	11.01%	11	3.37%	<0.001
Alcohol							<0.001
No	419	41.61%	182	26.73%	237	72.70%	
Light	495	49.16%	408	59.91%	87	26.69%	
Moderate	76	7.55%	74	10.87%	2	0.61%	
High	17	1.69%	17	2.50%	0	0.00%	
Drink alcohol (yes)	588	58.39%	499	73.27%	89	27.30%	<0.001
Hypertension	627	62.26%	410	60.21%	217	66.56%	0.051
Diabetes mellitus							0.203
No	818	81.23%	543	79.74%	275	84.36%	
Type 1	6	0.60%	5	0.73%	1	0.31%	
Type 2	183	18.17%	133	19.53%	50	15.34%	
Diabetes mellitus (yes)	189	18.77%	138	20.26%	51	15.64%	0.079
Dyslipidemia	487	48.36%	333	48.90%	154	47.24%	0.622

LA: left atrium; LVEF: left ventricular ejection fraction; HF: heart failure; HTN: arterial hypertension; COPD: chronic obstructive pulmonary disease; AF: atrial fibrillation.

**Table 10 jcm-10-03846-t010:** EQ-5D questionnaire.

**Variables**	**All**		**Men**		**Women**		***p*** **-Value**
EQ-5D questionnaire	941	93.45	639	93.83	302	92.64	0.474
Mobility	941		639		302		<0.001
I have no problems walking	648	68.86%	489	76.53%	159	52.65%	
I have some problemswalking	288	30.61%	149	23.32%	139	46.03%	
I have to stay in bed	5	0.53%	1	0.16%	4	1.32%	
Personal care	941		639		302		<0.001
I have no problems with personal care	862	91.60%	599	93.74%	263	87.09%	
I have some problems washing or dressing	73	7.76%	39	6.1%	34	11.26%	
I am unable to wash or dress myself	6	0.64%	1	0.16%	5	1.66%	
Daily life activities	941		639		302		<0.001
I have no problems performing daily life activities	722	76.73%	520	81.38%	202	66.89%	
I have some problems performing daily life activities	202	21.47%	114	17.84%	88	29.14%	
I am unable to perform daily life activities	17	1.81%	5	0.78%	12	3.97%	
Pain/discomfort	941		639		302		<0.001
I have no pain or discomfort	652	69.29%	485	75.9%	167	55.3%	
I have moderate pain or discomfort	262	27.84%	139	21.75%	123	40.73%	
I have much pain or discomfort	27	2.87%	15	2.35%	12	3.97%	
Anxiety/depression	941		639		302		<0.001
I am not anxious or depressed	655	69.61%	484	75.74%	171	56.62%	
I am moderately anxious or depressed	254	26.99%	137	21.44%	117	38.74%	
I am very anxious or depressed	32	3.40%	18	2.82%	14	4.64%	
EQ-5D index (summarizing score) *	941	0.82 ± 0.21	639	0.85 ± 0.19	302	0.74 ± 0.22	<0.001

* EQ-5D index (summarizing score): The steps described in the article by Herdman M., Badia X. and Berra S. (2001) were followed to calculate the index value of any health condition. EuroQol-5D: a simple alternative for assessing health-related quality of life in primary care. Atención Primaria, 28(6), 425–429.

**Table 11 jcm-10-03846-t011:** ACTS scale.

ACTS Questionnaire							
Variables	All		Men		Women		*p*-Value
Burden scale	738	53.43 ± 7.95	493	53.98 ± 8.02	245	52.31 ± 7.71	< 0.001
Benefit scale	738	11.34 ± 2.55	493	11.38 ± 2.62	245	11.26 ± 2.42	0.389
General burden scale	738	1.69 ± 1.01	493	1.67 ± 1.02	245	1.74 ± 1	0.224
General benefit scale	738	3.63 ± 1.01	493	3.62 ± 1.04	245	3.65 ± 0.95	0.980

## Data Availability

Data supporting the reported results can be found in a dataset of REGUEIFA Study (Odds S.L. A Coruña Spain) and can be consulted upon request.

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
