# Peer review of "Differences in the Clinical Profile and Management of Atrial Fibrillation According to Gender. Results of the REgistro GallEgo Intercéntrico de Fibrilación Auricular (REGUEIFA) Trial"

_jcm, 2021, doi:10.3390/jcm10173846_

Round 1

Reviewer 1 Report

Dear Pf. Mary Li,

It is my pleasure to review the manuscript entitled “Differences in the clinical profile and management of atrial fi-brillation according to gender. Results of the REGUEIFA (REg-istro GallEgo Intercéntrico de Fibrilación Auricular) trial”. The authors report a gender difference in AF patients undergoing ablation procedure. According to the authors, rhythm control measures were less frequently prescribed in women than in men, despite the fact that the former had poorer perceived quality of life and were more often symptomatic.

The study is well written and merits publication. However, there are several issues to be addressed.

#1. Did all patients with AF in 8 hospitals of Galician Health Department? Or there were exclusions due to patient refusal, missing data, drop outs, or screening failure? It would be helpful to describe how many patients were screened between 2 January 2018 and 27 February 2020 in 8 hospitals of the Galician Health Department.

#2. The authors should first describe full term for AHT in the abstract and main document. Furthermore, AHT is unusual acronym. How about just “hypertension” or “HTN”?

#3. Does FC (frequency control) refers to rate control?

#4. In Table 3, what does IAD refer to?

#5. Throughout the tables, the authors should describe “%” whenever indicated. It is confusing whether a given number is a percentage or actual number.

#6. In Table 6 with regard to EHRA symptom classification, can authors say women are more symptomatic? P value is 0.052. Also in first row of the Table 10, EQ-5D score did not differ between two sexes. And can the authors explain the difference between EQ-5D index (last row of Table 10) and EQ-5D score (first row)?

#7. In Table 6, “paroxysmal AF” is duplicated in “clinical type of AF” section.

#8. In Table 6, the authors should report the number of patients who underwent ablation and not number of patients who did not undergo ablation.

#9. In Table 7, the difference between men and women with regard to rhythm control strategy, the main driver was electrical cardioversion. This should be described in the discussion section.

#10. There are way too many tables… Please combine some of them and rearrange some of them to supplementary tables. However, if the Journal’s standards allow more than 10 tables, it will be good as it is.

#11. Any potential explanations for lesser rhythm control efforts for women? It is the core of this study.

#12. Please describe that this specific study was approved by relevant institutional review board.

Author Response

Point 1. Did all patients with AF in 8 hospitals of Galician Health Department? Or there were exclusions due to patient refusal, missing data, drop outs, or screening failure? It would be helpful to describe how many patients were screened between 2 January 2018 and 27 February 2020 in 8 hospitals of the Galician Health Department.

Response 1: The patients included in our study are those who agree to participate, and sign the informed consent, in eight Cardiology services in Galicia consecutively. Patients who refused to participate were not included in the study. In our article we describe the baseline data. In the future, the data obtained with the follow-up of the patients will be analyzed.

Point 2. The authors should first describe full term for AHT in the abstract and main document. Furthermore,AHT is unusual acronym. How about just “hypertension” or “HTN”?

Response 2: AHT is the acronym for arterial hypertension, but if it is considered unusual we have replaced it with HTN

Point 3. Does FC (frequency control) refers to rate control?

Response 3: FC, frequency control refers to rate control. We have replace frequency control with rate control. We have used RC as the acronym of rate control, and RhC as the acronym of rhythm control.

Point 4. In Table 3, what does IAD refer to?

Response 4: IAD refers to implantable automatic defibrillator. We have replaced it with ICD (implantable cardioverter defibrillator).

Point 5. Throughout the tables, the authors should describe “%” whenever indicated. It is confusing whether a given number is a percentage or actual number.

Response 5: We have added % to describe percentages.

Point 6. In Table 6 with regard to EHRA symptom classification, can authors say women are more symptomatic? P value is 0.052. Also in first row of the Table 10, EQ-5D score did not differ between two sexes. And can the authors explain the difference between EQ-5D index (last row of Table 10) and EQ-5D score (first row)?

Response 6: Statistical significance is certainly not reached (p = 0.052); However, it is observed that a higher percentage of men are in class I (36.42%) compared to 28.2% of women; and a higher percentage of women are in class III (11.35%) compared to 8.52% of men.

In table 10 first p-value refers to the number of patients, men and women, who answered the questionnaire; we can say a similar percentage of women (92,64%) and men (93,83%) answered the EQ-5D questionnaire.

We observed a estadistically significant differences (p<0,001) between genders in all terms of the questionnaire: mobility, personal care, daily life activities, pain/discofort and anxiety/depression.

So, we can say that the women with atrial fibrillation in our study have a poorer quality of life related to the arrhythmia evaluated according to a validated scale.

The EQ-5D is a generic health-related quality of life measurement instrument. The patient himself assesses his state of health, first in severity levels by dimensions (descriptive system) and then in a more general evaluation visual analog scale. A third element of the EQ-5D is the index of social values obtained for each state of health generated by the instrument.

The descriptive system contains five health dimensions (mobility, self-care, daily activities, pain / discomfort, and anxiety / depression) and each has three levels of severity (no problems, some problems or moderate problems, and serious problems). In this part of the questionnaire, the individual must mark the level of severity corresponding to their health status in each of the dimensions, referring to the same day that the questionnaire is completed. In each dimension of the EQ-5D, the severity levels are coded with a 1 if the answer option is "I have no problems"; with a 2 if the answer option is "some or moderate problems"; and with a 3 if the answer option is "many problems". The combination of the values of all the dimensions generates 5-digit numbers, with 243 possible combinations of health states, which can be used as profiles. The EQ 5D index of preference values for each health state is obtained from studies in the general population or in groups of patients in which several of the health states generated by the EQ-5D are assessed using an assessment technique such as time trade-off. The index ranges from 1 (best health) to 0 (death). In this way, an index is available that can be used directly. To calculate the value of any health status, first, the value of 1 is assigned to status 11111 (no health problems in any dimension). If the state is different from 11111, the value of the constant is subtracted (table 1). Later, if there are level 2 problems in a certain dimension, the value corresponding to each dimension is subtracted. The same procedure is followed when there are level 3 problems, although previously multiplying the value of the dimension with problems by 2. Finally, the coefficient that corresponds to the parameter N3 - a parameter that represents the importance given to level 3 problems in any dimension - it is subtracted only once when there is at least one dimension with level 3 problems. For example, in the case of health status 13111, the value 1 would be used and the constant y 0.2024 (0.1012 * 2) for having level 3 problems in the personal care dimension (table 1). In addition, the parameter N3 would be subtracted, which would finally give an index of 0.4355 (0.4355 = 1 0.1502 0.2024 0.2119).

Table 1. Coefficients for the calculation of the social tariff for the EQ 5D in Spain

Parameters Coefficients

Constant    0,1502

Mobidity     0,0897

Self-care     0,1012

Daily activities 0,0551

Pain/discomfort 0,0596

Anxiety/depression 0,0512

N3               0,2119

Adapted from Badia et al 1999

Point 7. In Table 6, “paroxysmal AF” is duplicated in “clinical type of AF” section.

Response 7: This point has been solved.

Point 8. In Table 6, the authors should report the number of patients who underwent ablation and not number of patients who did not undergo ablation.

Response 8: “No” refers to Number (N.º). This point has been solved.

Point 9. In Table 7, the difference between men and women with regard to rhythm control strategy, the main driver was electrical cardioversion. This should be described in the discussion section.

Response 9: As we described in te discussion section, “a total of 55,8% of the women in our study were subjected to rhythm control”, versus 66,6% of the men, with stadistically significan differences in the electrical cardioversion rate, which proved lower in women (27,3% versus 18,4%, p= 0,002)”.

Point 10. There are way too many tables… Please combine some of them and rearrange some of them to supplementary tables. However, if the Journal’s standards allow more than 10 tables, it will be good as it is.

Response 10: We think that tables can provide relevant information. Since they can be provided as supplementary information, we would like to keep them.

Point 11. Any potential explanations for lesser rhythm control efforts for women? It is the core of this study.

Response 11: Although this is a observational study, we speculate about posible causes in the discussion.

Point 12. Please describe that this specific study was approved by relevant institutional review board.

Response 12: We have provided an Ethical statement, were we indicate our study has been aproved by the Clinical Research Ethics Comitee of Galicia

Reviewer 2 Report

Thank you for this submission.The present study submitted by Durán-Bobín et al. investigates clinical profile and treatment of patients with atrial fibrillation (AF) stratified by gender. The authors conclude, women suffering from AF were older, more often symptomatic and presented with higher risk of both thrombembolic and bleeding complications. The study is well-conducted, the paper is interesting and methods are appropriate.

Comments:

  • Table 1: p values should be presented seperated by RC and FC (2 p values for each group comparison).
  • How many patients underwent coronary angiogram? Did this affect outcomes?
  • Kaplan-meier curves should be given for time to thrombembolic and bleeding events.
  • Please also provide a multivariable cox regression analysis, to decrease the chance of selection bias.
  • Did mortality rates differ among men and women? What were predictors of mortality in this study cohoirt?

Author Response

Point 1: Table 1: p values should be presented seperated by RC and FC (2 p values for each group

comparison):

Response 1: In relation with this analysis, we have been calculated the globally differences between gender groups independently of the rhythm or rate control strategy. The presentation in the table 1shows the subgropups, but p values represent the global group comparison (women vs men).

Point 2: How many patients underwent coronary angiogram? Did this affect outcomes?

Response 2: We have not reported this data, since we believe that it does not provide relevant information in relation to the subject of our study. This is an observacional cohort study, so we do not have follow-up data, or events.

Point 3: Kaplan-meier curves should be given for time to thrombembolic and bleeding events.

Response 3: In the same way, this is an observacional cohort study, so we do not have follow-up data, or events. At this point we can not offer kaplan-meier curves.

Point 4: Please also provide a multivariable cox regression analysis, to decrease the chance of selection bias.

Response 4: This is an observacional cohort study, so we do not have follow-up data, or events. At this point we can not provide regression analisys.

Point 5: Did mortality rates differ among men and women? What were predictors of mortality in this study cohoirt?

Response 5: This is an observacional cohort study, so we can′t provide follow-up data, or events data.

Round 2

Reviewer 1 Report

The authors have significantly improved their manuscript. 

I recommend acceptance. 

Thank you. 

Reviewer 2 Report

Quality of the manuscript has significantly improved. No further comments.